# Interannual Response of Reef Islands to Climate-Driven Variations in Water Level and Wave Climate

**Michael V. W. Cuttler** [1,*], **Kilian Vos** [2], **Paul Branson** [1,3], **Jeff E. Hansen** [1,4], **Michael O'Leary** [4], **Nicola K. Browne** [5] **and Ryan J. Lowe** [1,4,6]

1  Oceans Graduate School and UWA Oceans Institute, The University of Western Australia, Crawley, WA 6009, Australia; paul.branson@uwa.edu.au (P.B.); jeff.hansen@uwa.edu.au (J.E.H.); ryan.lowe@uwa.edu.au (R.J.L.)

2  Water Research Laboratory, School of Civil and Environmental Engineering, UNSW Sydney, Manly Vale, NSW 2093, Australia; k.vos@unsw.edu.au

3  Oceans and Atmosphere, CSIRO, Crawley, WA 6009, Australia

4  School of Earth Sciences, The University of Western Australia, Crawley, WA 6009, Australia; mick.oleary@uwa.edu.au

5  School of Molecular and Life Sciences, Curtin University, Bentley, WA 6102, Australia; nicola.browne@curtin.edu.au

6  ARC Centre of Excellence for Coral Reef Studies, The University of Western Australia, Crawley, WA 6009, Australia

*  Correspondence: michael.cuttler@uwa.edu.au

**Abstract:** Coral reef islands are among the most vulnerable landforms to climate change. However, our understanding of their morphodynamics at intermediate (seasonal to interannual) timescales remains poor, limiting our ability to forecast how they will evolve in the future. Here, we applied a semi-automated shoreline detection technique (CoastSat.islands) to 20 years of publicly available satellite imagery to investigate the evolution of a group of reef islands located in the eastern Indian Ocean. At interannual timescales, island changes were characterized by the cyclical re-organization of island shorelines in response to the variability in water levels and wave conditions. Interannual variability in forcing parameters was driven by El Niño Southern Oscillation (ENSO) cycles, causing prolonged changes to water levels and wave conditions that established new equilibrium island morphologies. Our results present a new opportunity to measure intermediate temporal scale changes in island morphology that can complement existing short-term (weekly to seasonal) and long-term (decadal) understanding of reef island evolution.

**Keywords:** reef islands; ENSO; CoastSat; shoreline variability; satellite-derived shorelines

---

## 1. Introduction

Reef islands are unconsolidated accumulations of biogenic (reef-derived) sediment that are typically low-lying (<5 m above mean sea level). These landforms face multiple threats from global climate change including sea level rise and changing wave climates as well as ocean warming and acidification. For example, sea level rise is likely to increase the frequency and severity of flooding events and island over-topping [1]. In addition, reef degradation due to ocean warming and ocean acidification may further threaten island stability by both reducing sediment supply and decreasing the reef's capacity to dissipate wave energy, resulting in increased energy at the shoreline [2,3]. Despite a recent focus on predicting future changes to reef islands, there remain contrasting predictions for

their future. On the one hand, some studies suggest that sea level rise will directly translate to island erosion, inundation, and groundwater contamination, thereby reducing the habitability of reef islands in the future [4,5]. In contrast, other studies suggest that some islands have the capacity for vertical accretion and horizontal expansion that may provide alternative adaptation pathways to future climate change [6–8].

Reef island formation and evolution is governed by the interaction between incident waves and water levels with reef-building organisms and the reef structure. Thus, forecasting the response of reef island morphology to future climate change requires a detailed understanding of how both the incident forcing conditions and the reef ecosystem will change [3]. Extensive work on reef hydrodynamics has shown that future sea level rise and increased reef degradation are expected to increase wave energy at the shoreline due to the decreased wave dissipation by the reef [5,9–13]; and that sea level rise will alter cross- and alongshore gradients in hydrodynamics, potentially resulting in a decreased sediment supply to island shorelines (i.e., erosion) [14,15]. These studies typically combine high temporal resolution, in situ observations of hydrodynamics (waves, water levels, circulation) with numerical models and sediment transport formulations to predict coastal response to observed (or modeled) changes in hydrodynamics. However, these studies have often lacked observations of shoreline dynamics at the corresponding timescales on interest, and, therefore, our understanding of the response of reef island morphodynamics to future oceanographic changes remains unclear.

Previous investigations into reef island morphodynamics have been conducted at either short (event to seasonal; e.g., [16,17]) or long (multi-decadal; e.g., [6,18–20]) timescales. Importantly, different shoreline proxies and measurement techniques have been used depending on the investigation's temporal scale of interest. Event to seasonal dynamics have primarily utilized high precision GPS surveys to map the "toe of beach", or the intersection of island sediments with the reef platform. This shoreline proxy represents the seaward extent of the island, and similar to typical sandy beaches, is highly variable in response to changing hydrodynamic conditions. By combining measurements of the toe of beach with in situ hydrodynamic measurements, prior studies have measured reef island shoreline response to weekly wind and wave variability, seasonal shifts in wind and wave climate, and extreme events (tropical cyclones and tsunamis) [16,17,21]. More recently, short-term studies have incorporated three-dimensional survey techniques including unmanned aerial vehicle (UAV) photogrammetry and LiDAR to investigate volumetric and height variability over seasonal to annual timescales [22–24]. Although these studies have highlighted the variable responses (both erosion and accretion) that island shorelines can experience at these relatively short timescales, the techniques used are often expensive or difficult to implement at regular intervals (e.g., monthly or seasonal) for extended periods of time (e.g., multiple years). Thus, there is limited information regarding how these short-term morphodynamics translate to longer-term variability.

At longer timescales (e.g., decadal), studies have primarily relied on remotely sensed imagery (satellite or aerial imagery) to track multi-decadal (40- to 70-year time periods) island planform evolution [19,25–29]. The imagery used is often high spatial resolution (sub-meter) imagery but collected infrequently and/or irregularly. Thus, to enable measurement of a consistent shoreline proxy across multi-decadal timescales and remove potential for misinterpretation of shoreline change [19,30], these studies have mostly tracked the vegetation line. The vegetation line is typically slow to respond to oceanographic conditions and, as such, the high-frequency oscillations captured by the toe of beach are filtered out [18,19]. Through tracking the vegetation line, decadal-scale studies have shown diverse patterns of island evolution, including increases in planform area despite long-term sea-level rise [6,20]. However, due to the sparse temporal resolution of these observations (e.g., less than 10 images across 40+ years) [26], there is limited ability to resolve interannual island morphodynamics or the roles of different physical processes (waves, water levels) operating over a range of timescales (seasonal to decadal) in driving the observed long-term changes. Thus, there remains a gap in our understanding of the interannual variability of reef island dynamics which is needed to understand how short-term

dynamics translate to long-term changes and the roles of different physical processes in driving longer-term changes.

Recently, there has been increased focus internationally on using publicly available, medium spatial resolution (10–30 m), but high temporal resolution (weekly to monthly) satellite imagery to assess shoreline changes [31,32]. These approaches present a new opportunity to enhance our understanding of the intermediate timescale (seasonal to interannual) dynamics of reef islands by providing novel temporal resolution observations (5–16 days imagery for 30+ years), which can provide a more detailed understanding of processes driving shoreline change and enhance our ability to forecast the response reef island shorelines to future changes. Here, we use a modified version of CoastSat [31,33] to identify the response of reef island morphodynamics to key morphological drivers at interannual timescales (1999–2019). Our results highlight significant interannual variability in island morphology in response to sustained changes in wave conditions and water levels that were primarily driven by large-scale climate fluctuations. More generally, these results provide insights into how islands may respond to future climate shifts and present a new opportunity to link short-term, process-based studies with longer-term geomorphological observations.

## 2. Materials and Methods

### 2.1. Study Area

Our study area is the Pilbara region along the northwest of Western Australia (WA), which hosts Australia's largest reef island archipelago with over 50 reef-fringed sand islands spread over ~250 km between Exmouth and Dampier (Figure 1). This study focused on four inshore islands located at the southwest end of the archipelago in Exmouth Gulf: Eva, Y, Fly, and Observation Islands (Figure 1b). The region has a seasonally variable wave climate that fluctuates between Southern Ocean swell that enters Exmouth Gulf from the NW in winter, and local wind-generated waves from the SW in summer (Figure 1c,d). The Pilbara is also one of the most cyclone-prone regions globally, experiencing ~3 tropical cyclones per year [34–36].

More broadly, the WA coastline experiences relatively large seasonal and interannual variability in waves, wind, and water levels (Figure 2) that vary in response to larger-scale climate variability (e.g., El Niño Southern Oscillation, Ningaloo Niño) [37–41]. These four islands were selected as a test case for our new method due to the availability of wave and water level data, the availability of a topo-bathymetric LiDAR dataset [22] that was used for error assessments, and because they experience a seasonal variation in wind and wave climate that is common to many Indo-Pacific reef islands [17,30]. Furthermore, the interannual variability common to the WA coastline presents the opportunity to observe how short-term morphodynamics correspond to longer-term variability in waves and water levels. Finally, the Pilbara Islands are uninhabited and without any coastal infrastructure and, thus, present an opportunity for investigating "natural" island morphodynamics across seasonal to interannual timescales without human interventions.

### 2.2. Satellite-Derived Shorelines

To obtain our high temporal resolution dataset, reef island shorelines were detected from publicly available satellite imagery (Landsat-7, Landsat-8, Sentinel-2) using a modified version of the CoastSat workflow [33]. CoastSat is a Python-based toolkit that enables semi-automated detection of shorelines from publicly available satellite imagery. Our modified version of CoastSat—"CoastSat.islands"—is publicly available at https://github.com/mcuttler/CoastSat.islands. Here, we describe the modifications we made to the original CoastSat algorithm; further details of the underlying sub-pixel detection technique are found in Vos et al. [31,33].

Satellite-derived shorelines were obtained from July 1999 to July 2019 to coincide with the time period covered by the closest available tide gauge (Exmouth tide gauge, ~40 km West of Eva Island; Figure 1). Landsat-7, Landsat-8 (Tier 1 Top-of-Atmosphere), and Sentinel-2 (Level 1-C) images covering



the four reef islands during the study period were accessed with Google Earth Engine and then analyzed using the CoastSat.islands workflow. The original CoastSat algorithm uses two steps to map the shoreline (defined as the instantaneous interface between sand and water) on the satellite images: (1) pixel-wise image classification using a two-layer neural network and (2) thresholding of the sand–water interface using the Modified Normalized Difference Water Index (MNDWI) and extraction of the contour at sub-pixel resolution with the Marching Squares algorithm [42]. However, since this technique was developed for open coast sandy beaches, the presence of shallow submerged reef flats around the islands' shoreline was found to cause a significant seaward bias in the satellite shorelines and, therefore, two major modifications were implemented. First, the image classification scheme was re-trained using manually digitized pixels from images exclusively of the Pilbara reef islands. Second, instead of thresholding the MNDWI image, the island contour was obtained directly from the classified image as the sub-pixel resolution boundary between "sand" and "water" pixels (Figure 3). We use the "regionprops" function of the scikit-image package [43] to measure island orientation directly from the classified image; the island contour was used to calculate shoreline position and sub-aerial island area (see below).

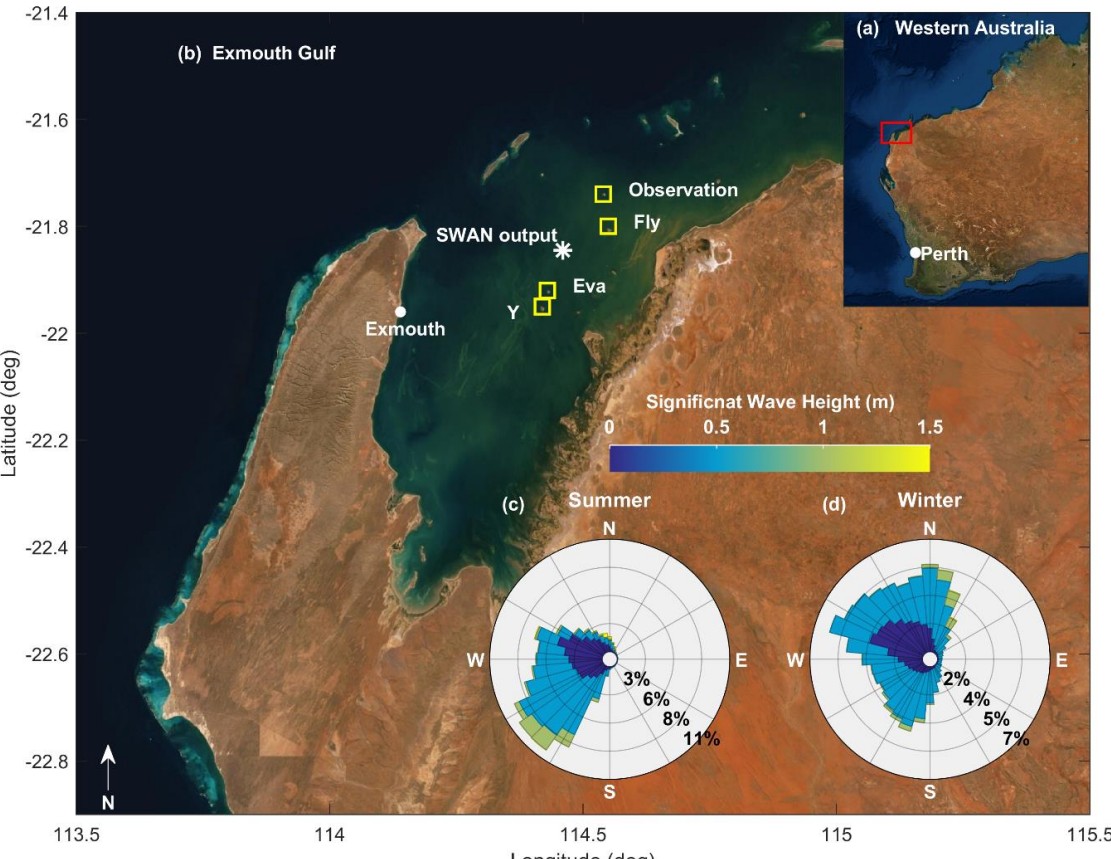

**Figure 1.** (**a**) Location of the study area, Exmouth Gulf, in the southwestern Pilbara (red box). (**b**) Satellite image of Exmouth Gulf including the location of the reef islands examined here (yellow squares), SWAN wave model output location (white asterisk), and location of the Exmouth tide gauge (white dot). (**c**,**d**) Characteristic wave climate for summer (**c**) and winter (**d**) in Exmouth Gulf. Percentages indicate frequency of occurrence for waves from a particular direction, and the colors indicate significant wave height. Wave parameters were extracted from a SWAN hindcast at the location marked in (**b**) (white asterisk). Basemap images in (**a**,**b**) are from ESRI Online.

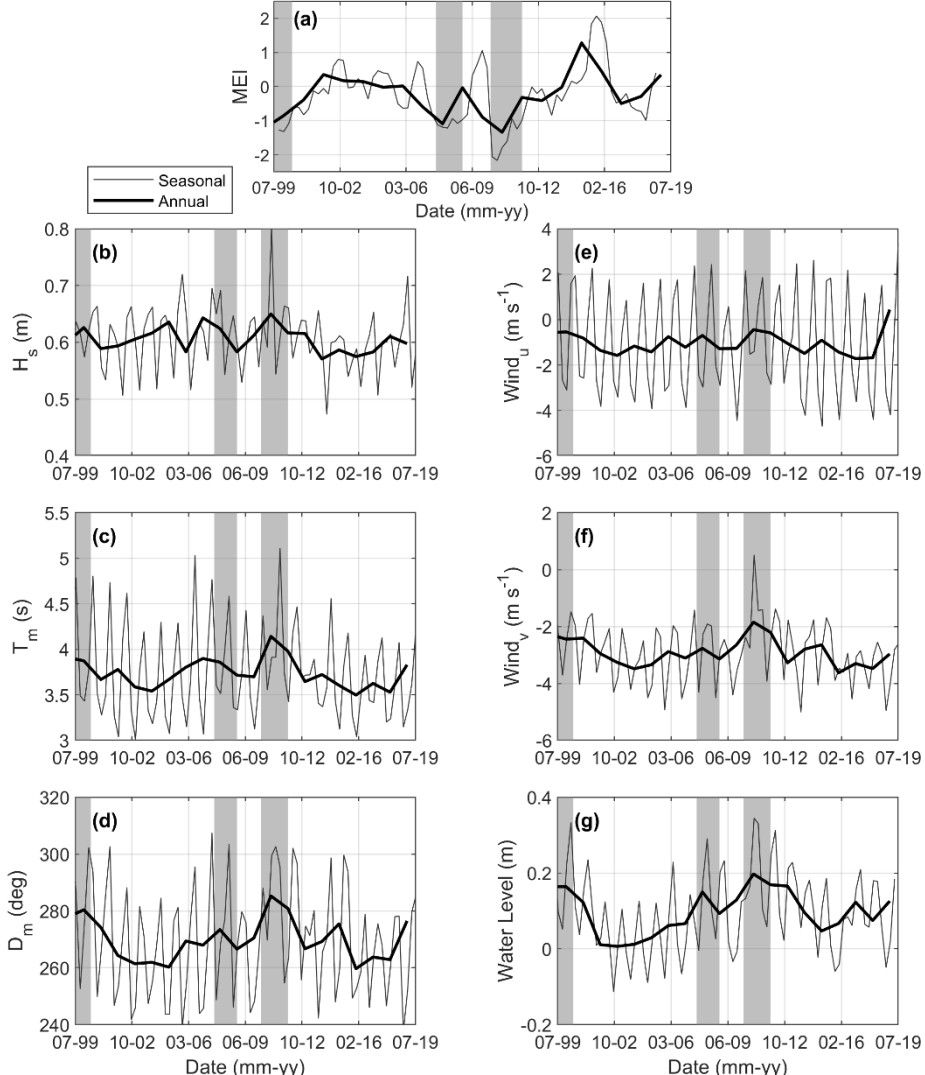

**Figure 2.** Seasonal (dark grey) and annual averaged (black line): (**a**) Multivariate El Niño Southern Oscillation (ENSO) Index (MEI), (**b**) significant wave height, (**c**) mean wave period, (**d**) mean wave direction, (**e**) easterly wind component, (**f**) northerly wind component, and (**g**) water level in Exmouth Gulf from 1999–2019. Wave and wind variables (**b**–**f**) were derived from our hindcast model (white asterisk in Figure 1); water level data are from the Exmouth tide gauge maintained by the Western Australia Department of Transport. In all panels, grey shading denotes La Niña events, classified as periods when MEI was less than −1.

As part of our post-processing, Landsat-7 and Landsat-8 images that had positional uncertainty greater than 10 m were removed from the dataset. Similarly, Sentinel-2 images flagged by the data provider (European Space Agency) as of insufficient geometric quality (there was no positional accuracy data provided for Sentinel-2 imagery) were also discarded (on average 7% of raw data removed). Since the satellite images are acquired at different stages of the tide, a linear tidal correction was applied to the satellite shorelines. The instantaneous island contours were translated horizontally in the shore-normal direction to a reference elevation of 0.75 m Australian height datum (AHD), which corresponds to mean high water (MHW), using the measured tidal elevation at the Exmouth tide gauge (~40 km from the islands, see Figure 1) and a linear inter-tidal beach slope obtained from a 2018 topo-bathymetric LiDAR survey [22]. Tide-corrected island polygons were utilized to compute time-series data for a range of island metrics including shoreline position and sub-aerial sand area. Similar to other shoreline change methods (e.g., the Digital Shoreline Analysis System [44,45]),

shoreline changes through time were assessed along 25 m alongshore-spaced cross-shore transects. Although we assumed a constant beach slope through time, previous tests using a time-varying slope showed no significant improvement in the accuracy associated with the CoastSat workflow [31].

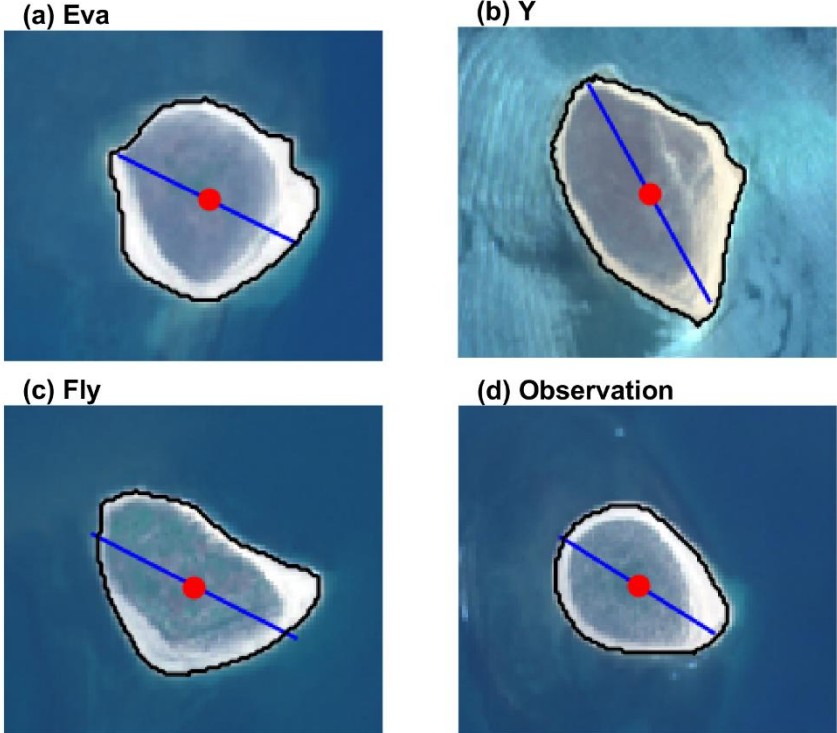

**Figure 3.** Example CoastSat.islands detection showing the detected shoreline (black line), centroid of the sandy area (red dot), and the major axis orientation (blue line) at the (**a**) Eva, (**b**) Y, (**c**) Fly, and (**d**) Observation islands.

CoastSat has been extensively validated using several sites from a diverse range of open coast sandy beaches and has been shown to be accurate to within 10–15 m [31]; however, as this is an application in a new environment (i.e., reef-fronted shorelines) we endeavored to estimate the horizontal error in our shoreline positions. Therefore, we directly compared our satellite-derived shorelines and planform metrics to a topo-bathymetric LiDAR dataset collected over these islands from 11 October 2018 [22]. The MHW shoreline contour was extracted at each island from the LiDAR dataset using ArcMap (version 10.6) and used to calculate LiDAR-derived horizontal shoreline position (using the same alongshore transects as used in CoastSat) and planform area and orientation. For the area and orientation comparison, LiDAR-derived and satellite-derived metrics were directly compared for each satellite. For the shoreline position comparison, we calculated the difference in relative horizontal shoreline position (relative to the long-term average shoreline position) at each alongshore transect between the LiDAR-derived and satellite-derived shoreline.

### 2.3. Wave and Water-Level Data

To understand the relationship between met-ocean conditions and the observed shoreline dynamics, we combined in situ water level observations (Exmouth tide gauge, Figure 1b) with a 20 year regional-scale, fully coupled wave-circulation hindcast (Delft3D-SWAN [46,47]) that has been extensively validated along the Pilbara coast [48]. Delft3D-SWAN coupled models have been used successfully in reef environments for simulating the effects of tropical cyclones [49] as well as ambient conditions [50]. The model extended 500 km in the alongshore direction from Exmouth Gulf in the west to the Dampier Archipelago in the east, and approximately 100 km offshore to the 200 m

depth contour (Supplementary Materials Figure S1). A Delft3D hydrodynamic model was run on a three-dimensional, curvilinear grid with 10 sigma layers and a resolution of approximately 1500 m at the boundary to 150 m in the inshore region of the west Pilbara coast. The SWAN wave model was run on a set of three nested grids with resolution varying from 2500 m offshore to 100 m inshore in the proximity of the islands. Model bathymetry was derived from a digital elevation model made available through the Western Australian Marine Science Institution (WAMSI) Dredging Science Node [48] and augmented with the 2018 topo-bathymetric LiDAR data obtained for the immediate region around the study islands.

The forcing for surface (10 m elevation) winds and mean sea-level pressure was provided by the European Centre for Medium-Range Weather Forecasts ERA5 data-assimilating hindcast [51]. The ocean boundary of the wave model was forced with spatially variable frequency and directional spectra from the Centre for Australian Weather and Climate Research (CAWCR) Wave Hindcast at an hourly time interval [52], and the hydrodynamic model was forced with the Oregon State University inverse tide model (v8) [53] with 13 constituents (M2, S2, N2, K2, K1, O1, P1, Q1, M4, MM, MF, MS4, and MN4). The Delft3D–SWAN model was validated against in situ wave gauges that were deployed at Eva, Fly, and Observation islands between April 2018 and November 2018 (Supplementary Materials Figure S1). All modeled wave parameters (i.e., significant wave height, mean wave period) showed moderate agreement with in situ wave measurements across the validation period (average root mean square error 0.25 m and 2.75 s for wave height and period, respectively; Supplementary Materials Figure S1). This is likely related to the fine scale bathymetric variability around the islands that is not fully resolved by the finest-scale wave model (~100 m resolution). However, as our aim was to investigate the larger-scale hydrodynamic conditions, as opposed to the reef-scale dynamics at each island, these results provide sufficient confidence in applying this model.

Water-level observations (relative to the Australian height datum, AHD) from July 1999 to July 2019 were obtained from the Exmouth tide gauge provided by the Western Australia Department of Transport (DoT). Climate variability was assessed using the Multivariate ENSO Index (MEI), which was downloaded from the National Oceanic and Atmospheric Administration (NOAA) Physical Sciences Laboratory (https://psl.noaa.gov/enso/mei/). For this analysis, an MEI less than −1 corresponded to La Niña conditions, and an MEI greater than 1 corresponded to El Niño conditions.

### 2.4. Data Processing

The sampling frequency of the satellite observations was dictated by each satellite's revisit period (16 days for Landsat-7 and Landsat-8 and approximately 5 days for Sentinel-2) and the availability of cloud-free images. As the focus of the present work was seasonal to interannual morphodynamic variability, the time-series of shoreline changes were monthly averaged (when more than one shoreline per month existed) and any small data gaps were filled using linear interpolation. Long-term shoreline change trends were assessed by calculating a weighted linear regression rate of shoreline change [44,45] using the "raw" (not-averaged) shoreline position time-series, where the weighting was calculated as $1/e^2$ and $e$ was the root mean square positional error determined for each satellite at each island (i.e., values in Figure 4).

Physical and climatological variables (e.g., winds, waves, water levels, MEI) were averaged from hourly to monthly intervals in order to assess their role in driving the observed shoreline changes. Model output wave parameters from a representative location in Exmouth Gulf (~12 m depth) (Figure 1) were used to assess wave climate variability based on the significant wave height ($H_s$), mean wave period ($T_m$), and mean wave direction ($D_m$).

All monthly data were temporally averaged to seasonal and annual timescales, with seasons defined as: summer (December, January, February); autumn (March, April, May); winter (June, July, August); spring (September, October, November). Annual anomalies for environmental forcing data were calculated as the difference between the annual average and the 20 year average.

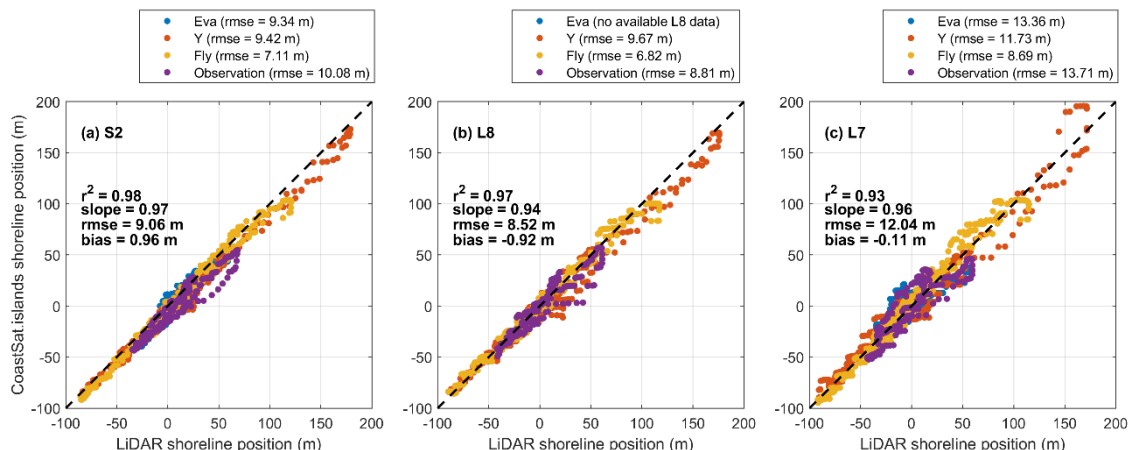

**Figure 4.** Validation of the (**a**) Sentinel-2, (**b**) Landsat-8, and (**c**) Landsat-7 satellite-derived shorelines versus airborne LiDAR survey. Total statistics (reported in black text) were calculated on the combined datasets (all islands) after the average shoreline position for each transect at each island was removed. Island-specific statistics are reported within the legend above each panel. In all panels, blue denotes Eva, orange denotes Y, yellow denotes Fly, and purple denotes Observation.

## 3. Results

### 3.1. Accuracy of Satellite-Derived Shorelines

Comparisons between the tide-corrected shoreline positions and the 2018 airborne LiDAR survey yielded an average root mean square error (RMSE) of 9.06 m, 8.52 m, and 12.04 m for shorelines derived from Sentinel-2 (S2), Landsat-8 (L8), and Landsat-7 (L7), respectively (Figure 4). Accuracy varied among the four islands and ranged from 7.11 to 10.08 m for S2, from 6.82 to 9.67 m for L8, and from 8.69 to 13.71 m for L7. These results are in agreement with the horizontal accuracy reported by previous studies that employed Landsat and Sentinel-2 imagery to map changes in shoreline position [32,54,55], including the more general application of CoastSat along five diverse open coast sandy beaches [31]. When comparing the satellite-derived island areas to LiDAR-derived area, an RMSE of 10,626 $m^2$ (~1 hectare) was observed (Supplementary Materials Figure S2), which is approximately 3% of the 20 year average area for Y and Fly and 6% of the 20 year average area for Eva and Observation. The RMSE for island orientation was 10.5 deg (Supplementary Materials Figure S2); however, this is biased by a relatively large error for the orientation derived from Landsat-7 for Eva (blue triangle in Supplementary Materials Figure S2). This error is related to an offshore bias in the Landsat-7 shoreline position in the northwest of Eva that skews the satellite-derived orientation. If this point is removed, the RMSE for orientation reduces to 2.6 deg. As the analysis below relies on monthly, seasonal, and annual averaging, the shoreline variability is significantly reduced (i.e., smoothed) and, thus, the averaged values reported below may be within the calculated error limits but are still considered measurable changes.

### 3.2. Variability of Reef Island Shorelines

Our shoreline dataset consisted of 565 to 650 shorelines per island between 1999 and 2019 (~29 per year), which is a two-order of magnitude increase in observational resolution compared to the sample frequency available from historical aerial or higher resolution satellite imagery [20,26]. Over the 20 year study period (1999–2019), all islands exhibited highly dynamic, cyclical shoreline and planform geometry variability (Figure 5). Measured island areas ranged from ~15,000 $m^2$ to ~18,000 $m^2$, ~30,000 $m^2$ to ~37,000 $m^2$, ~24,000 $m^2$ to ~29,000 $m^2$, and ~14,000 $m^2$ to ~17,000 $m^2$ for Eva, Y, Fly, and Observation, respectively. Island orientation showed large fluctuations at Eva (~100 degrees) and more moderate changes at Fly (~20 degrees) and Observation (~10 degrees), whereas Y only varied by ~4 degrees over the 20 year period. Across this 20 year period, the islands showed alongshore-variable

shoreline change trends. Eva exhibited long-term accretion along an SW–NE axis, and erosion along a NW to SE axis (Figure 5). The other three islands were characterized by adjacent erosion/accretion sections focused on their SE sections where sand spits occur (Figure 5).

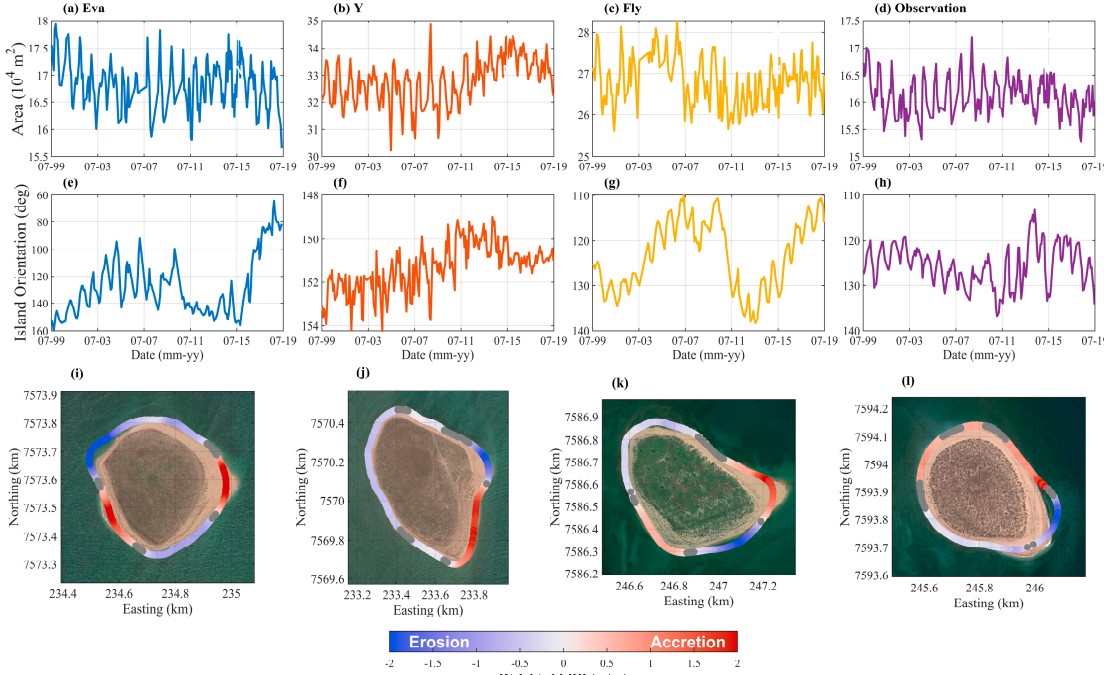

**Figure 5.** (**a**–**d**) Temporal variability of reef island area (monthly averaged); note different *y*-axis limits. (**e**–**h**) Temporal variability or major axis orientation (monthly averaged); note different *y*-axis limits. (**i**–**l**) Weighted linear regression rate (LRR) of change from 1999–2019 at each alongshore transect. Blue denotes an erosive trend, red denotes an accreting trend, and grey denotes no significant trend (assessed at a 95% confidence level).

The islands showed seasonal variability that persisted across the 20 year study period (Figure 6). These fluctuations corresponded to the changing position of the sand spits on the eastern halves of all islands (Supplementary Materials Figure S3) and accounted for shoreline changes ~±20 m, area changes ~±5000 m$^2$, and orientation fluctuations ~±5 deg (Figure 6; Supplementary Materials Figure S4). In general, these seasonal island adjustments were season-specific and coherent amongst the four islands with all islands showing a more E–W orientation during summer–autumn and more N–S orientations during winter–spring (Supplementary Materials Figures S5–S8). Similarly, the island planform area was smallest during the austral autumn and winter (March–August) and largest during the austral spring and summer (September–February), corresponding to the highest and lowest seasonal water levels, respectively (Supplementary Materials Figures S5–S8).

The high-frequency (seasonal) variability was superimposed on longer-term interannual variability regarding the 20 year average shoreline position and planform geometry (Figure 6). Across all islands, the interannual signal was characterized by alongshore alternating regions of erosion and accretion that translated to interannual variability in island orientation (Figure 6). The interannual variability occurred in multi-year phases (~3–5 years) that corresponded to distinct island geometry (Figures 6 and 7). For example, at the start of the record (1999–2002), Fly showed accreted shorelines along the southeastern sections and eroded along their eastern–northeastern and southern–southwestern sections (Figure 6c). This pattern then reversed in 2005–2010, when the SE shorelines were eroded, but the ENE and SSW shorelines were accreted. From 2011–2015, the shoreline positions returned to their 1999–2002 states, and from 2015 to present, shorelines have re-aligned to their 2005–2010 states (Figure 6c,e, yellow line).

Comparable dynamics were observed at Eva and Observation, whereas at Y, only one reversal was observed, occurring in 2011 (Figure 6).

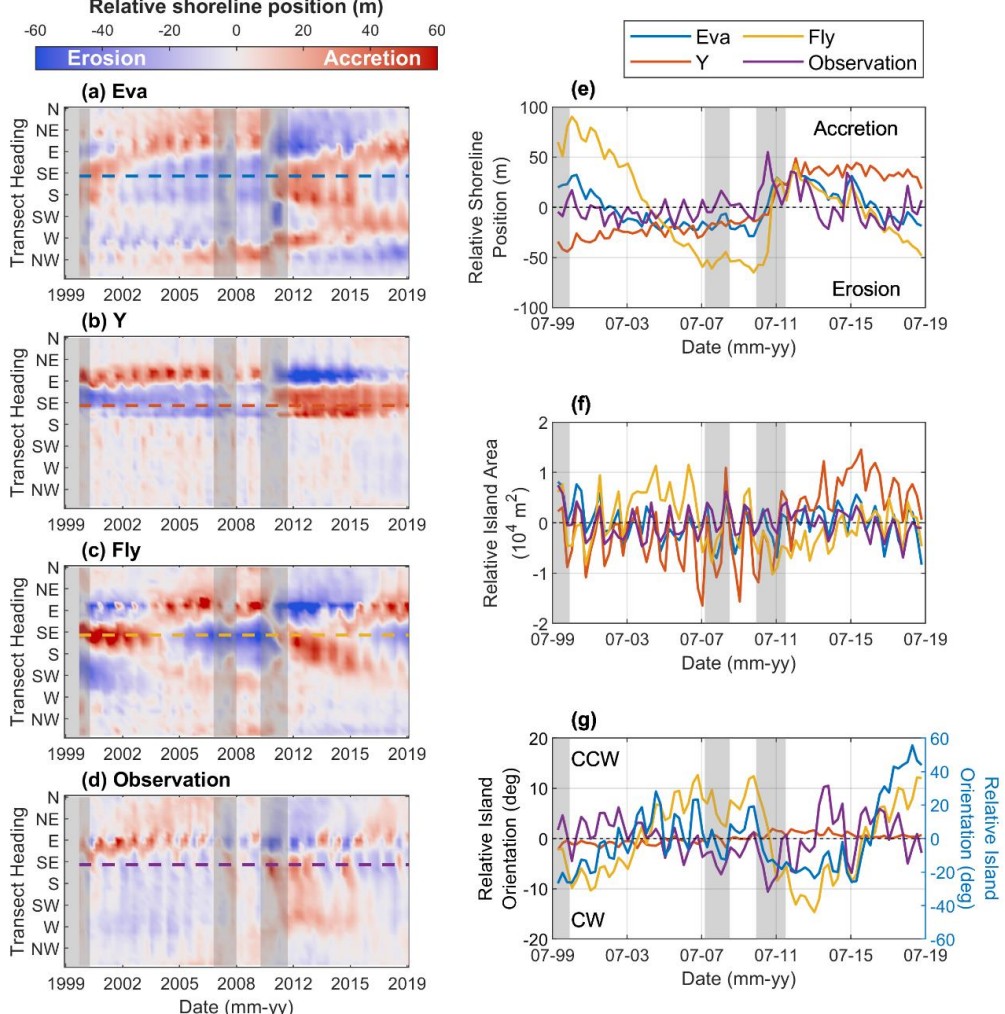

**Figure 6.** (**a**–**d**) Seasonally averaged shoreline position (referenced to the 20 year mean shoreline position) at (**a**) Eva, (**b**) Y, (**c**) Fly, and (**d**) Observation islands. Blue denotes erosion and red denotes accretion. (**e**–**g**) Seasonally averaged (**e**) shoreline position at a representative transect from the southeastern (~140° from north, dashed line in panels **a**–**d**) side, where positive indicates accretion and negative indicates erosion, (**f**) island area (relative to 20 year average area), and orientation (relative to 20 year average orientation), where positive denotes counter-clockwise (CCW) rotation and negative denotes clockwise (CW) rotation. As the rotation of Eva Island was greater than others, there is a right-hand axis for Eva Island (blue) and a left-hand axis for Y, Fly, and Observation islands. The 140° transect was used, as it is indicative of shoreline changes across all four islands. In panels (**e**–**g**), the blue line represents Eva Island, the orange line is Y Island, the yellow line is Fly Island, and the purple is Observation Island. In all panels, grey shading denotes La Niña events, classified as periods when MEI is less than −1.

### 3.3. Drivers of Island Shoreline Variability

Over seasonal timescales, the study region experiences a fluctuation between local, wind-generated waves during austral summer (characterized by short-period waves from the southwest) and Southern Ocean swells in austral winter (characterized by long-period waves from the west–northwest) (Figure 1). The water level is highest in austral autumn and lowest in austral spring, corresponding to the seasonal cycle of water levels in eastern Indian Ocean [37] (Figure 2). The seasonal variability in wave and

water level conditions drive alternating erosion–accretion patterns for the northern and southern side of the islands. For example, as the seasonal wave direction becomes more northerly in winter, northern shorelines erode and the southern shorelines accrete, and vice versa as waves become more southerly in summer (Figure 2).

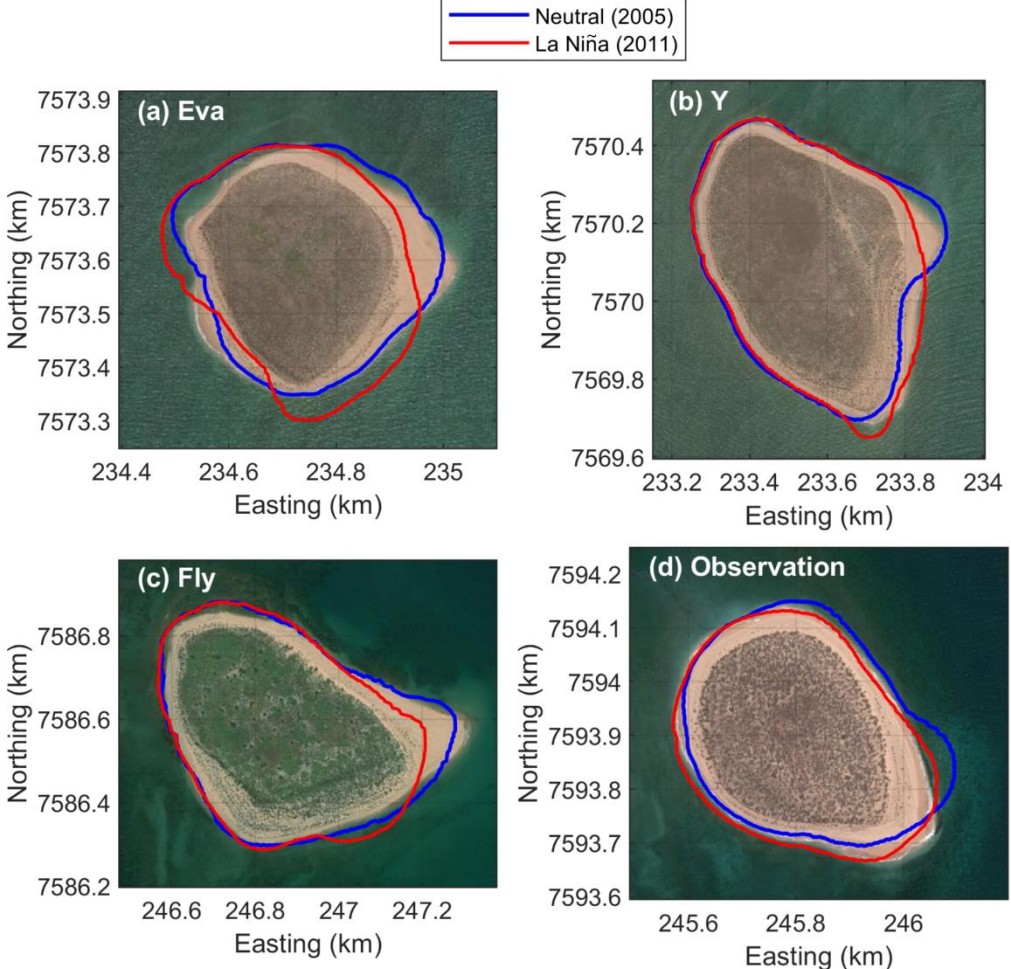

**Figure 7.** Interannual shoreline variability: (**a**–**d**) annually averaged shoreline position at each island for 2005 (blue line), an ENSO neutral year; and 2011 (red line), a strong La Niña year.

The interannual variability in met-ocean conditions was characterized by distinct periods of anomalous waves, winds, and water levels (Figure 8). For example, in 1999 and 2011, there were sustained increases in water levels (0.05–0.10 m), northerly winds (0.75–1 m/s), mean wave period (0.2–0.5 s), and mean wave direction (10–20 degrees more northerly) (Figure 8). Both of these periods corresponded to La Niña events, defined as periods when the Multivariate ENSO Index (MEI) [39] was less than −1 (Figure 8). Although our time series only partially resolved the 1998–2001 La Niña event, we fully resolved the 2010–2012 event, which was the strongest La Niña event in recent times [56]. During the 2010–2012 La Niña event, the Pilbara coast experienced sustained increases in water levels (0.1 m) and changed wave conditions (15° clockwise rotations in wave direction, 0.05 m increase in significant wave height, 0.5 s increase in wave period) (Figure 8). The anomalous met-ocean conditions were significantly correlated with ENSO variability at the 95% confidence level, with La Niña causing higher mean water levels ($r = -0.74$, $p \ll 0.01$), increased wave height ($r = -0.69$, $p \ll 0.01$), and wave period ($r = -0.65$, $p \ll 0.01$) and more northerly waves ($r = -0.55$, $p \ll 0.01$). These ENSO events modified the regional wind, wave, and water level conditions for several consecutive years, leading to a re-adjustment of the island morphology, including persistent erosion of northern shorelines and

accretion of southern shorelines (~40 m) and clockwise island rotations (~10°) (Figure 8). There was a weak La Niña in 2007, but this did not drive similar anomalies in wind, waves, and water levels and, thus, there was no observed reversal in island shoreline position (Figure 8).

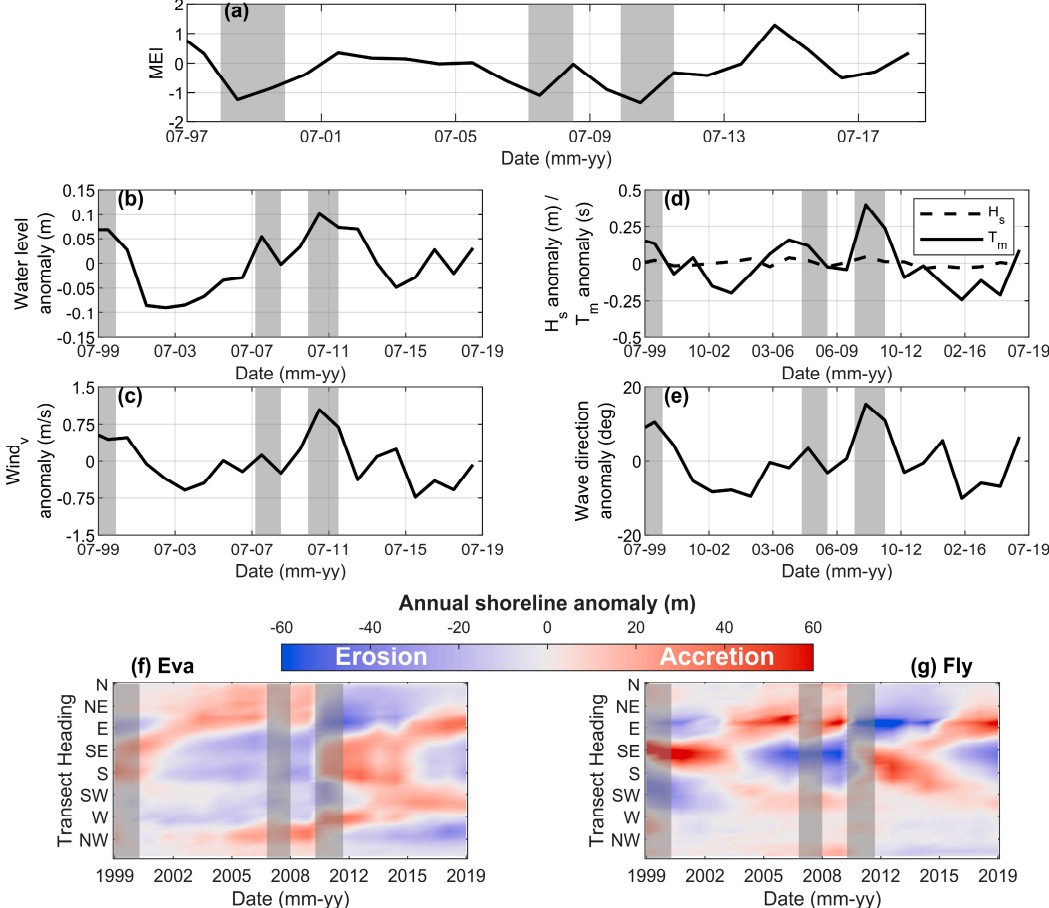

**Figure 8.** (**a**) Multivariate ENSO Index (MEI), where values less than −1 denote La Niña. (**b**–**e**) Interannual anomalies for (**b**) total water level, (**c**) southerly component of wind velocity, (**d**) mean wave period (black line) and significant wave height (dashed black line), and (**e**) mean wave direction. (**f**,**g**) Interannual shoreline variability at (**f**) Eva and (**g**) Fly, where blue shading denotes erosion, and red shading denotes accretion. In all panels, grey bars denote La Niña events.

## 4. Discussion

### 4.1. Variability of Island Shorelines and Relationship to Climate Fluctuations

The morphology of reef islands is related to the location of wave convergence zones (or "nodal points") that are created by wave refraction around the islands [16,57,58]. The location of convergence zones varies with changing met-ocean conditions, for example, changes in wave direction due to the seasonal wave climate variability or due to the water level fluctuations altering wave refraction over the reefs [16,17,57]. Our results show similar island fluctuations that are related to the seasonal variability in wind, wave, and water-level climates as would be predicted by previous studies. However, key to this study is that we are able to capture how these dynamics evolve over interannual timescales. The seasonal variability exists as high-frequency oscillations about the longer-term (interannual) signal (Figure 6, Supplementary Material Figure S4). This suggests that the interannual met-ocean conditions set an "equilibrium" island morphology and that the seasonal variability consists of fluctuations about this interannual "equilibrium" morphology. For example, from 2011–2013, the islands show sustained N–S orientations (~10 deg clockwise rotations) in response to increased northwesterly waves

associated with La Niña. Yet, the typical seasonal cycle observed here still persists, albeit around a new (more N–S) equilibrium orientation (Figure 6; Supplementary Materials Figure S4). Given that the interannual variability observed here was directly linked to ENSO variability, the "equilibrium" morphologies, corresponded to different ENSO states with neutral/El Niño corresponding to accretion on the north, erosion on the south, and La Niña corresponding to erosion on the north, accretion on the south (Figures 7 and 8), which translates to extended periods of time (several years) where entire sections of island shorelines are in a more (or less) eroded state.

This interannual to ENSO-driven changes in waves, wind, and water levels, were similar to previous observations at open (no reef) sandy coasts [59,60]; however, this has yet to be directly shown for reef islands. Interestingly, although ENSO events may subside rapidly (i.e., within a year), there is a lag in island response whereby after a La Niña event (e.g., 1999, 2011) islands do not return to their modal planform state for several years (Figures 6 and 8). Further, our results suggest that islands can rapidly adjust their planform geometry to sustained changes in met-ocean conditions. This is similar to recent studies that have suggested that islands can adapt to future sea level rise through vertical accretion [7,8]. However, as our results are purely two-dimensional, we were unable to resolve whether the observed changes were due to the new sediment transported across reef platforms to the shoreline or whether shoreline extension was due to the enhanced foredune erosion [24]. For example, given that these islands are relatively high (average 8 m elevation, [22]) sustained periods of higher water levels and altered wave conditions under La Niña could have increased foredune erosion yielding more sub-aerial sand at specific alongshore regions, or the increased water levels could have enabled more wave energy to penetrate reef flats and alter cross-shore sediment transport [14,15].

The tropical Indo-Pacific (where most reef islands are located) experiences ENSO-driven, interannual variability in wind, waves, and water levels [59,61,62]. Although climate fluctuations have been suggested to be important for island dynamics [63,64], their role has yet to be fully assessed due to the fact of our limited ability to monitor reef island morphodynamics over interannual timescales. Our high temporal resolution dataset suggests that ENSO-related processes (and potentially other similar climate fluctuations) are drivers of interannual reef island variability more broadly and should be considered when assessing future adjustments to climate change. Furthermore, the natural interannual variability in water levels (0.10 m) and wave conditions (15° change in mean wave direction) of this region are comparable to changes in wave conditions and water levels forecasted to occur along many coastlines in the next several decades [65,66]. Therefore, these islands can provide insight into the potential response of reef islands more broadly to sustained, multi-year changes in waves and water levels.

## 4.2. Applicability of Satellite-Derived Shorelines for Reef Islands

Previous investigations into reef island morphodynamics have relied on short-term, high accuracy (order 1 m or better), three-dimensional measurements (e.g., GPS surveys, UAV, LiDAR) [16,22–24] or long-term (decadal) high spatial resolution (order meters or better) aerial or satellite imagery [18–20,27,67,68]. Three-dimensional approaches have the benefit of being able to capture volumetric and elevation changes, which can be critical for determining island sensitivity to short-term fluctuations in boundary conditions. However, they are often expensive and, therefore, difficult to implement at a high temporal frequency and/or large spatial extent. Similarly, high spatial resolution imagery can provide accurate assessments of decadal-scale planform change across large spatial scales, but is often only available sporadically (i.e., ~5 time points across 30–70 years; [26]). Importantly, limited temporal resolution (at both short and long timescales) can also alias morphodynamic variability at intermediate (interannual) timescales. However, for islands that experience shoreline changes ~10–15 m or larger, our approach can complement existing methods by filling the temporal gap between short-term (seasonal) and long-term (decadal) studies, and resolving interannual shoreline dynamics.

Our dataset utilized satellite-derived shorelines from medium spatial resolution (10–30 m resolution) and high temporal resolution (5–16 days) satellite imagery. This yielded several hundred

shorelines per island that spanned 20 years, which corresponds to a two-order of magnitude increase in observations compared to previous investigations [26]. The temporal frequency of our observations allowed us to observe seasonal to interannual morphodynamic variability. Our method relies on several assumptions that are based on the data availability and should be considered when applying this technique to other islands. Firstly, a time-invariant beach slope was used to tidally correct the instantaneous shorelines across the 20 year study period. It is likely that the beach slope around the island varied over time. However, although there are methods available for determining a time-averaged beach slope [69], as well as methods for estimating time-varying beach slope from high-frequency (hourly, daily) video imagery [70–72], there are presently no methods that the authors are aware of for estimating a time-varying beach slope from satellite imagery across 20+ years. In addition, previous validation of the CoastSat workflow with a time-varying tidal-correction showed no indication of a time-dependent error that could be attributed to the variability in beach slope [31]. Secondly, accurately georeferenced (<10 m) Landsat and Sentinel-2 imagery is not available everywhere on the globe and, in certain locations, co-registration of the images may be required prior to shoreline extraction. Finally, the neural network that underlies the classification was trained on imagery from the Pilbara; this could be re-trained (https://github.com/kvos/CoastSat) using imagery from any other reef island to optimize the detection.

## 5. Conclusions

Reef islands are some of the most dynamic landforms globally but also some of the most vulnerable to climate change. To better predict island responses to future changes, we need to enhance our understanding of island morphodynamics across seasonal to interannual timescales. Here, we utilized publicly available satellite imagery to track interannual island morphodynamics from 1999–2019. Our results reveal that interannual variability in shoreline behavior is related to sustained changes in waves, wind, and water-level conditions triggered by ENSO cycles. Islands demonstrated planform geometries that were characteristic of specific ENSO states (e.g., La Niña or neutral/El Niño). The relationship between island morphology and ENSO highlights a key timescale of variability that should be considered when assessing reef island vulnerability and provides new insight into island response to interannual and longer-term climate shifts.

**Supplementary Materials:** The following are available online at http://www.mdpi.com/2072-4292/12/24/4089/s1, Figure S1: Model domains and validation, Figure S2: Comparison of LiDAR-derived and satellite-derived reef island area and orientation, Figure S3: Seasonal planform geometry; Figure S4: Seasonal and interannual variability of reef island morphology; Figure S5: Seasonal planform variability—Eva; Figure S6: Seasonal planform variability—Y; Figure S7: Seasonal planform variability—Fly; Figure S8: Seasonal planform variability—Observation.

**Author Contributions:** M.V.W.C. conceived the study, adapted the CoastSat algorithm, and analyzed all results. K.V. adapted the CoastSat algorithm and contributed to image analysis and processing. P.B. developed the numerical wave-circulation model and conducted the 20 year hindcast. M.O. contributed to study design and processed LiDAR data. J.E.H. contributed to study design, method development and data analysis and interpretation. N.K.B. contributed to study design and data interpretation. R.J.L. contributed to method development and interpretation of results. All authors wrote the manuscript. All authors have read and agreed to the published version of the manuscript.

**Funding:** This research was supported by the Australian Research Council Centre of Excellence for Coral Reef Studies (CE140100020). N.B. was supported by an Australian Research Council DECRA Fellowship (DE180100391). K.V. was supported by a UNSW Scientia PhD scholarship.

**Acknowledgments:** The authors would like to thank Airborne Research Australia for LiDAR data collection and the Western Australian Department of Transport tide data. We appreciate the efforts of the United States Geological Survey and the European Space Agency in providing publicly available satellite imagery to the scientific community and Google Earth Engine for facilitating access to the image archives. This work was supported by resources provided by the Pawsey Supercomputing Centre with funding from the Australian Government and the Government of Western Australia.

**Conflicts of Interest:** The authors declare no conflict of interest.

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
