# Peer review of "Interannual Response of Reef Islands to Climate-Driven Variations in Water Level and Wave Climate"

_remotesensing, doi:10.3390/rs12244089_

Round 1

Reviewer 1 Report

This paper deals with coral reef morphodynamics at intermediate (seasonal to interannual) timescales. They applied publicly available satellite imagery to investigate the evolution of a group of reef islands located in the eastern Indian Ocean. They reported interesting result regarding the relationship between the island morphology and ENSO. This paper’s results are interesting and thus suitable for publication in Journal of Marine Science and Engineering, MDPI. There are only small comments as follows.

  1. Line 171, Figure 3: What is the mathematical definition of the major axis? How to obtain its direction from the satellite image?
  2. What is a physical process from El Nino La Nina to cause morphology change in the study area?

Author Response

We thank the reviewer for their suggestions and comments on our manuscript. Incorpoation of these suggestions have greatly improved our manuscript. Please see attached for detailed response to each reviewer suggestion. 

Reviewer 2 Report

The authors present an interesting study on the interannual response of reef islands to variations in water level and wave climate. The analysed variations in foricng conditions serve as proxy to possible changes in climate.

The presented framework features innovative methods and obtained results are of interest for the readers.

Thus, my recommendation is for acceptance, with some minor comments/suggestions to the authors:

L173-174: "...images that had positional uncertainty greater than 10m were removed from the dataset....(L176)...were also discarted". Please provide the % of data removed from the dataset following this filtering criteria.

L246-247: Please rephrase or remove brakets.

L265: Clould you please specify here the RMSE on the island orientation. This would ocmplete the other two and put the variability results in context to the method error. Related to this, in general, resulting variabilities in the obtained trends are greater than the errors, and this should be highlighted in greater manner in the discussion.

L339: Figure 3: I think you mean FIgure 2.

Figure S4: If I understood correctly the graphs S4(a,b,c) have a different reference shoreline than those in Figure 6 in the main manuscript. if so, please specify in the figure caption the reference shoreline.

Author Response

We thank the reviewer for their helpful comments and suggestions, which have greatly improved our manuscript. Please see attached for detailed response to each reviewer suggestion. 

Author Response

We thank the reviewer for their detailed suggestions and comments which have greatly improved our manuscript. Please see attached for detailed response to each reviewer suggestion. 
